# Revealing mechanisms of infectious disease spread through empirical contact networks

**Pratha Sah**[1], **Michael Otterstatter**[2,3], **Stephan T. Leu**[4], **Sivan Leviyang**[5], **Shweta Bansal**[1]*

**1** Department of Biology, Georgetown University, Washington, District of Columbia, United States of America, **2** British Columbia Centre for Disease Control, Vancouver, Canada, **3** School of Population and Public Health, University of British Columbia, Vancouver, Canada, **4** School of Animal and Veterinary Sciences, The University of Adelaide, Roseworthy, Australia, **5** Department of Mathematics & Statistics, Georgetown University, Washington, District of Columbia, United States of America

* shweta.bansal@georgetown.edu

**Data Availability Statement:** All empirical datasets are available at: https://doi.org/10.7910/DVN/YAHRDJ The code for the INoDs software is

## Abstract

The spread of pathogens fundamentally depends on the underlying contacts between individuals. Modeling the dynamics of infectious disease spread through contact networks, however, can be challenging due to limited knowledge of how an infectious disease spreads and its transmission rate. We developed a novel statistical tool, INoDS (Identifying contact Networks of infectious Disease Spread) that estimates the transmission rate of an infectious disease outbreak, establishes epidemiological relevance of a contact network in explaining the observed pattern of infectious disease spread and enables model comparison between different contact network hypotheses. We show that our tool is robust to incomplete data and can be easily applied to datasets where infection timings of individuals are unknown. We tested the reliability of INoDS using simulation experiments of disease spread on a synthetic contact network and find that it is robust to incomplete data and is reliable under different settings of network dynamics and disease contagiousness compared with previous approaches. We demonstrate the applicability of our method in two host-pathogen systems: *Crithidia bombi* in bumblebee colonies and Salmonella in wild Australian sleepy lizard populations. INoDS thus provides a novel and reliable statistical tool for identifying transmission pathways of infectious disease spread. In addition, application of INoDS extends to understanding the spread of novel or emerging infectious disease, an alternative approach to laboratory transmission experiments, and overcoming common data-collection constraints.

## Author summary

Network models are widely used to understand and predict infectious disease spread in human and animal populations. However, the choice of network model often relies on subjective expert knowledge or disease transmission experiments that are time-consuming and difficult to perform. We developed a novel tool, called INoDS (Identifying contact Networks of infectious Disease Spread), that uses robust statistical approach to establish relevance of a network model in explaining transmission pathways of an infectious disease

available at: https://github.com/bansallab/INoDS-model.

**Funding:** SB and PS was supported by the National Science Foundation Ecology and Evolution of Infectious Diseases Grant 1216054 (https://www.nsf.gov/funding/pgm_summ.jsp?pims_id=5269). STL was supported by an Australian Research Council DECRA fellowship (DE170101132, https://www.arc.gov.au/grants/discovery-program/discovery-early-career-researcher-award-decra), and an ARC grant to CM Bull (DP130100145), which funded the sleepy lizard project (https://www.arc.gov.au/grants). The funders had no role in study design, data collection and analysis, decision to publish, or preparation of the manuscript.

**Competing interests:** The authors have declared that no competing interests exist.

outbreak. We used computer simulations and real-world dataset to test the accuracy of our tool and robustness to missing data. We found that INoDS is robust to common data-collection constraints, broadly applicable and accurate compared to current approaches. The tool that we have developed can therefore provide immediate epidemiological insights in the event of an epidemic outbreak, and can be used to improve targeted disease control.

## Introduction

Host contacts, whether direct or indirect, play a fundamental role in the spread of infectious diseases [1–4]. Traditional epidemiological models make assumptions of homogeneous social structure and mixing among hosts which can yield unreliable predictions of infectious disease spread [3, 5–7]. Network approaches provide an alternative to modeling infection transmission by explicitly incorporating host interactions that mediate pathogen transmission. Formally, in a contact network model, individuals are represented as nodes, and an edge between two nodes represents an interaction that has the potential to transmit infection. A dynamic contact network model tracks interactions evolving over time due to social, demographic or environmental processes as well as perturbations [8]. Constructing a complete contact network model requires (i) knowledge about the transmission route(s) of a pathogen, (ii) a sampling of all individuals in a population, and (iii) a sampling of all interactions among the sampled individuals that may lead to infection transfer. In addition, accuracy of disease predictions depends on the precise epidemiological knowledge about the pathogen, including the rate of pathogen transfer given a contact between two individuals, and the rate of recovery of infected individuals.

The use of modern technology in recent years, including RFID, GPS, radio tags, proximity loggers and automated video tracking has enabled the collection of detailed movement and contact data, making network modeling feasible. Despite the technology, logistical and financial constraints still prevent data collection on all individuals and their social contacts [9–14]. More importantly, limited knowledge about a host-pathogen system makes it challenging to identify the mode of infection transmission, define the relevant contacts between individuals that may lead to infection transfer, and measure the per-contact rate of infection transmission [15–17]. Laboratory techniques of unraveling transmission mechanisms usually take years to resolve [18–20]. Defining accurate contact networks underlying infection transmission in human infectious disease has been far from trivial [3, 21, 22]. For animal infectious disease, limited information on host behavior and the epidemiological characteristics of the spreading pathogen makes it particularly difficult to define a precise contact network, which has severely limited the scope of network modeling in animal and wildlife epidemiology [15, 23].

Lack of knowledge about disease transmission mechanisms has prompted the use of several indirect approaches to identify the link between social structure and disease spread. A popular approach has been to explore the association between social network position (usually quantified as network degree) of an individual and its risk of acquiring infection [24–27]. Another approach is to use proxy behaviors, such as movement, spatial proximity or home-range overlap, to measure direct and indirect contact networks occurring between individuals [28–30]. A recent approach, called the $k$-test procedure, explores a direct association between infectious disease spread and a contact network by comparing the number of infectious contacts of infected cases to that of uninfected cases [31]. However, several challenges remain in identifying the underlying contact networks of infection spread that are not addressed by these

approaches. First, it is often unclear how contact intensity (e.g. duration, frequency, distance) relates to the risk of infection transfer unless validated by transmission experiments [19]. Furthermore, the role of weak ties (i.e., low-intensity contacts) in pathogen transfer is ambiguous [21, 32]. The interaction network of any social group will appear as a fully connected network if monitored for a long period of time. As fully-connected contact networks rarely reflect the dynamics of infectious disease spread through a host population, one may ask whether weak ties can be ignored, or what constitutes an appropriate intensity threshold below which interactions are epidemiologically irrelevant? Second, many previous approaches ignore the dynamic nature of host contacts. The formation and dissolution of contacts over time is crucial in determining the order in which contacts occur, which in turn regulates the spread of infectious diseases through host networks [8, 33, 34]. Finally, none of the existing approaches allow direct comparison of competing hypotheses about disease transmission mechanisms which may generate distinct contact patterns and consequently different contact network models.

All of these challenges demand an approach that can allow direct comparison between competing hypothesis on transmission pathways while taking into account the dynamics of host interactions and constraints of data sampling. We introduce a statistical tool called INoDS (*Identifying contact Networks of infectious Disease Spread*) that establishes the epidemiological relevance of observed contact networks in explaining the patterns of infectious disease spread. INoDS also allows testing competing hypotheses on the mode of disease transmission by performing model comparison between different contact networks. The tool can estimate the per-contact transmission rate and recovery rate for various disease progressions (e.g. SI, SIS and SIR) and can be extended to incorporate complexities in transmission (e.g. individual-level heterogeneity in susceptibility; latent period following infection). INoDS provides inference on static and dynamic contact networks, and is robust to common forms of missing data. Using two empirical datasets, we highlight the two-fold application of INoDS—(*i*) to identify whether observed patterns of infectious disease spread are likely given an empirical contact network, and (*ii*) to identify transmission routes, the role of the contact intensity, and the per contact transmission rate of a host-pathogen system. The epidemiological insights into infectious disease provided by INoDS can be invaluable in implementing immediate disease control measures in the event of an emerging epidemic outbreak.

## Results

The primary purpose of INoDS is to evaluate whether an observed contact network is likely to generate an infection time-series observed in a particular host population. INoDS also provides epidemiological insights into the spreading pathogen by estimating the per-contact rate of transmission ($\beta$). The pattern of infectious disease spread in a host population depends on the mode of transmission, and the epidemiological relevance of contact network is sensitive to the amount of collected data on nodes and edges. The tool therefore treats each empirically collected contact network as a unique network hypothesis, and facilitates hypothesis testing between different contact networks.

The INoDS algorithm follows a three step procedure (Fig 1). First, the tool estimates a per-contact transmission rate ($\beta$) of the pathogen and an background transmission parameter ($\epsilon$). The $\beta$ parameter quantifies the per-contact rate of pathogen transmission, and the $\epsilon$ parameter quantifies components of infection transmission that are unexplained by the contact network (S1 Fig). In the second step, Bayesian hypothesis testing is performed to establish epidemiological relevance of the observed contact network. Null hypothesis is expressed as a uniform distribution over randomized networks generated by permuting 10%—100% edge connections of each time-slice using double-edge swap procedure [35] and assigning each edge the average

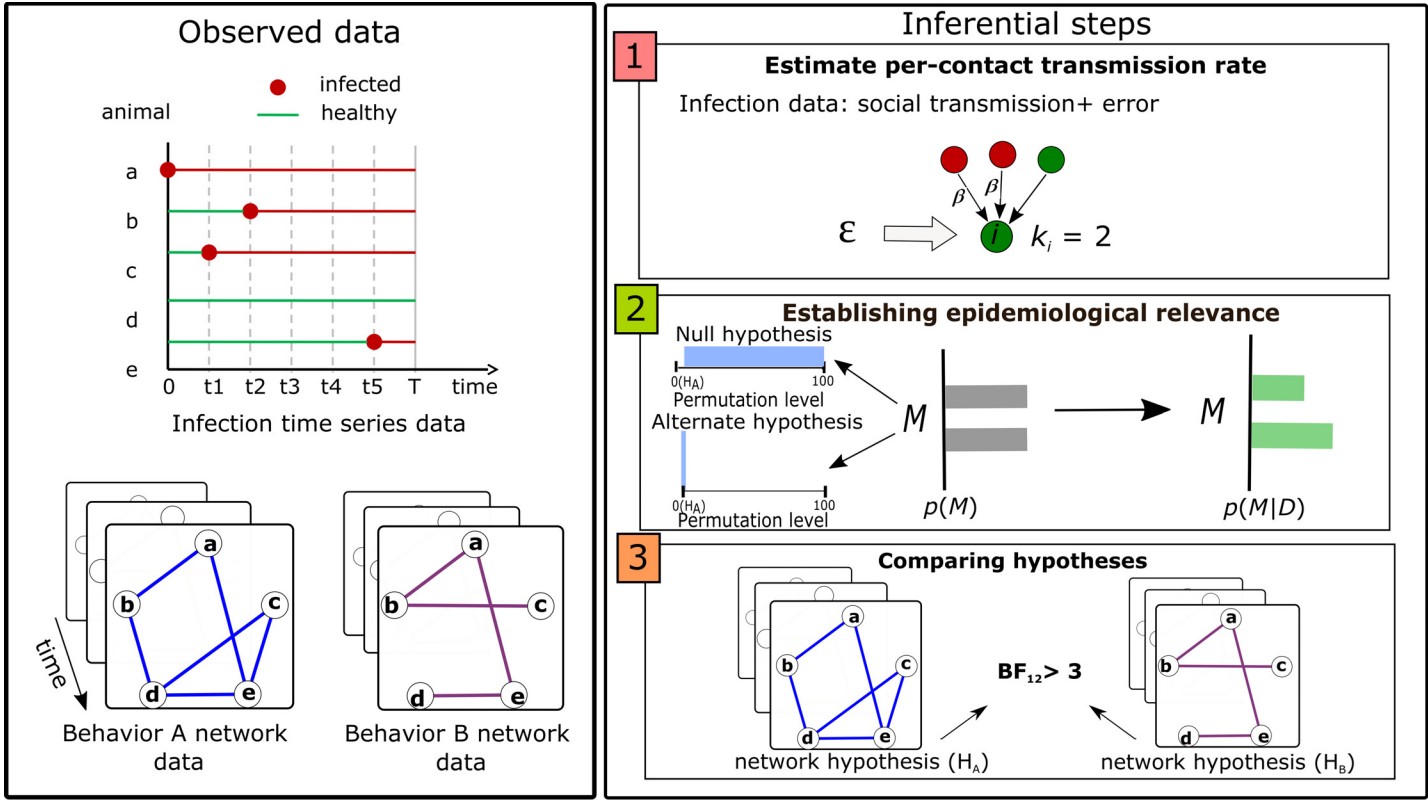

**Fig 1. A schematic of our algorithm. Observed data**(left panel): INoDS utilizes an observed infection time-series data to estimate statistical evidence towards a static or dynamic contact network hypothesis (or hypotheses) using a three-step procedure. Shown here is an example of two competing network hypotheses based on behaviors A and B that potentially cause infection transfer. **Inferential steps** (right panel): In the first step, the tool estimates per-contact transmission rate parameter $\beta$, and background transmission rate parameter $\epsilon$ which captures the components of infection propagation unexplained by the edge connections of the network hypothesis. Here, the total infected connections of the focal node $i$ ($k_i$) is 2. Second, to estimate the epidemiological relevance of the network hypothesis, Bayesian hypothesis testing is performed. The prior distribution shows that the null hypothesis ($M = 1$) assumes a uniform distribution over randomized networks generated by permuting 10%—100% of edge connections in the contact network ($H_A$), whereas the alternate hypothesis ($M = 2$) is a spike-shaped distribution such that only the contact network ($H_A$, 0% permutation) has non-zero probability. The distribution on model index shifts to $M = 2$ if the alternate hypothesis has higher posterior probability than the null. Third, model selection of competing network hypotheses is performed using Bayes Factor (BF). A Bayes factor above 2.44 is considered to be decisive support for one hypothesis over the other.

edge weight of the original snapshot. In the final step, model selection between multiple contact network hypotheses is performed using Bayes Factor.

In the sections that follow, we validate each of the three steps of INoDS, assess its robustness to missing data and compare its performance with previous approaches by performing simulation experiments of disease spread on a synthetic contact network. The advantage of using a simulated dataset is that it allows accurate evaluation of INoDS performance. We further demonstrate the applicability of INoDS in two empirical datasets: (*i*) spread of an intestinal pathogen in bumble bee colonies, and (*ii*) salmonella spread in Australian sleepy lizards.

## Validating INoDS performance

We validated the performance of INoDS using a simulated dataset. This dataset was generated by performing numerical disease simulations on a synthetic dynamic network (henceforth called the true synthetic network) where per-contact transmission rate, $\beta$, ranged from 0.01 to 0.1.

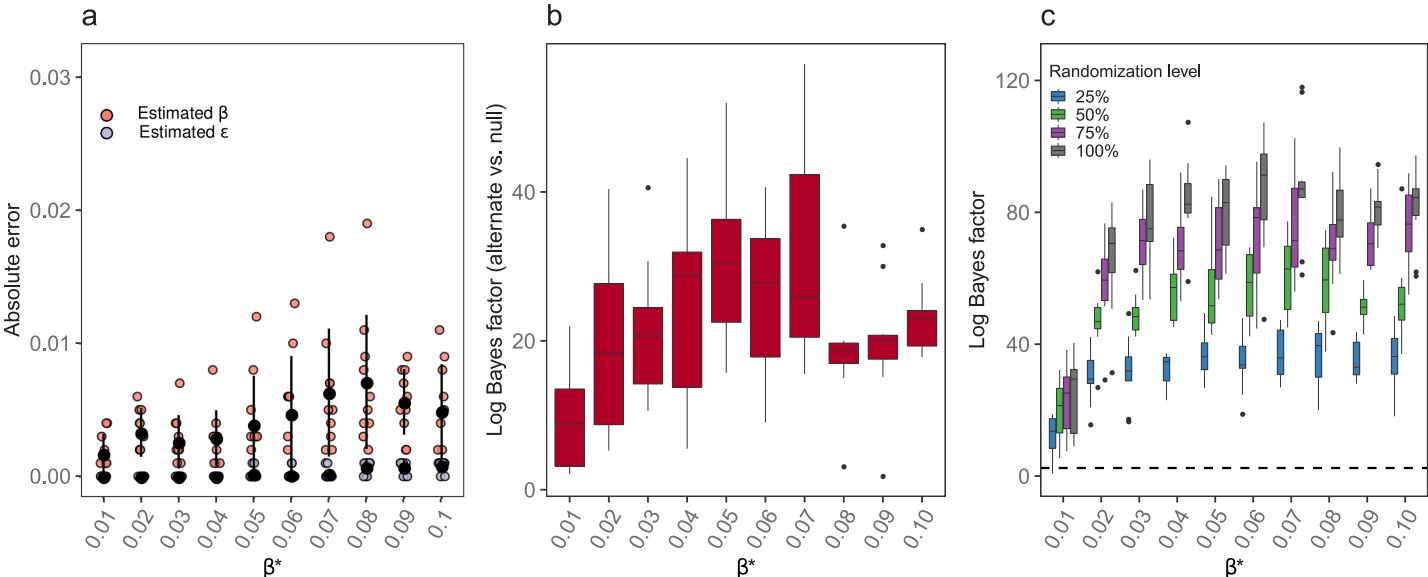

**Fig 2. Validation of the three steps of INoDS.** (a) **Step 1**: Absolute error in estimates of per-contact transmission rate parameter $\beta$ (orange circles) and background transmission rate $\epsilon$ (purple circles) for the simulated dataset with disease transmission rate ($\beta^*$) ranging from 0.01 to 0.1. The true value of background transmission rate ($\epsilon^*$) is zero. The filled black circle indicates the average absolute error and the error bars indicate standard deviation around the mean value. (b) **Step 2**: establishing epidemiological relevance of the observed contact network. Each box summarizes log Bayes factor of observed network compared to null hypothesis (viz a prior of networks with 10% to 100% permuted edges). (c) **Step 3**: model selection between the observed contact networks (0% randomization level) and networks with increasing edge randomization (25%, 50%, 75% and 100%). Log Bayes factor was calculated by substracting the log marginal evidence of randomized networks from log marginal evidence of the true (0% randomized) synthetic network. Log Bayes factor of more than 2.44 (dashed line) is considered to be a decisive evidence in favor of the observed contact network. The middle black line in each box plot is the median, the boxed area extends from the 25th to 75th quartile, and whiskers extended from the hinge to the largest/smallest value no further than 1.5 times the inter-quartile range.

**Validation of step 1**: Fig 2a shows INoDS estimates of $\beta$ and $\epsilon$ each for 10 independent disease simulations of the synthetic pathogen with disease contagiousness ranging from 0.01 to 0.1. We found that INoDS accurately estimated the per-contact transmission rate $\beta$, and background transmission rate $\epsilon$ for the simulated dataset. The accuracy was independent of the pathogen's contagiousness. For example, we estimated an average $\beta$ value of 0.039 (SD = 0.003) using INoDS for disease simulations involving a simulated pathogen with $\beta$ value of 0.04. The background transmission parameter, $\epsilon$, was accurately estimated as zero for all simulations since all infection transmission events in the simulated dataset were perfectly explained by the edge connections of the contact network (Fig 2a).

**Validation of step 2**: The second step, which involves establishing the epidemiological relevance of the contact network, was evaluated by performing Bayesian hypothesis testing. Prior was defined as an ensemble of randomized networks with 10%–100% permuted edge connections. We found that log Bayes factor of observed network was more than 2.44 for all ranges of $\beta$, indicating that INoDS accurately detected epidemiological relevance of the contact network irrespective of contagiousness of the spreading pathogen (Fig 2b).

**Validation of step 3**: Where multiple hypotheses of contact networks exists, model selection is performed by computing the ratio of marginal likelihoods (Bayesian evidence). We validated this step by comparing the Bayesian evidence of the true synthetic network with networks generated by shuffling 25%, 50%, 75% and 100% of edge connections present in the true synthetic network. We found that log Bayes factor of the true synthetic contact network was more than 2.44 for most replicates compared with randomized networks when $\beta^* = 0.01$. Log Bayes factor exceeded 10 for synthetic pathogens with transmission rates $\beta^* > 0.01$, suggesting a decisive evidence for the true synthetic network (Fig 2c).

## Robustness to missing network data

Next, we tested the robustness of INoDS against two potential sources of error in network data collection: incomplete sampling of individuals in a population (missing nodes) and incomplete sampling of interactions between individuals (missing edges). To create networks with missing data, we randomly deleted 25–75% of nodes and edges from the true synthetic network that were not a part of the path of simulated infection spread. We focus on this type of missing data as infected individuals and their contact are more likely to be observed (particularly for infections with observable symptoms). Methodologically, this approach also allows us to tease apart INoDS's performance when both the complete and incomplete networks have equal ability to explain the propagation of disease. If robust to missing data, we expect INoDS to recover the same parameter estimates and model evidence as the true and complete synthetic network.

We found that even when 75% of network data is missing, the estimated transmission rate $\beta$ is identical to $\beta$ values estimated for the complete synthetic network (Fig 3a). For all incompletely sampled networks, log Bayes factor was greater than 2.44 compared to the null hypothesis. INoDS thus correctly identified all incompletely sampled networks to be epidemiological relevant (Fig 3b). As expected, log Bayes factor was less than 2.44 for all degrees of missing data indicating that the true synthetic network does not have higher evidence compared with incompletely sampled networks. Together, we found that the performance of INoDS was unaffected by missing network data if the missing nodes/edges are not a part of the outbreak path.

In the supplement, we tested the performance of INoDS when data is missing at random for nodes, edges or cases (S3 Fig). We found that the deviation of estimated $\beta$ increases with the degree of missing data, and networks with missing data had lower evidence compared with the true synthetic network. This is because removing nodes/edges involved in the transmission process lowers the network's ability to explain the propagation of disease outbreak.

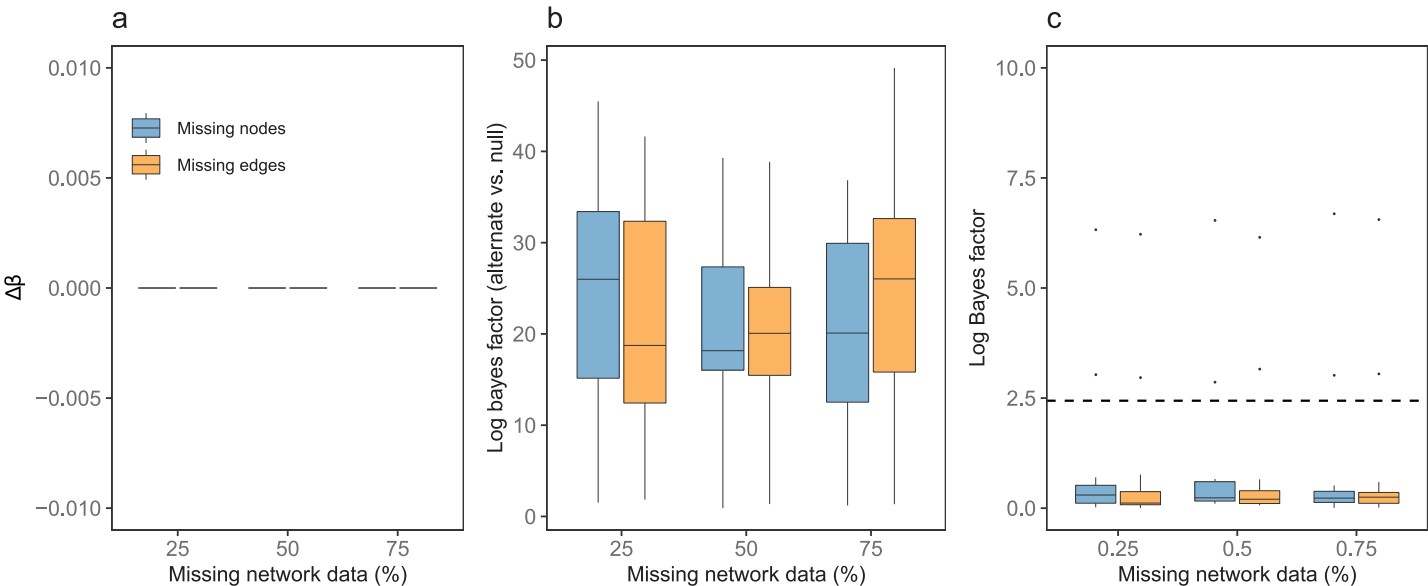

**Fig 3. Robustness of INoDS to missing network data.** Robustness of INoDS to missing nodes and missing edges in network hypothesis. Networks with missing nodes/edges were created by randomly removing 25–75% of nodes/edges not involved in infection spread path at each time-step from the dynamic synthetic network. (a) Step 1: $\Delta\beta$ is the relative deviation of estimated transmission parameter $\beta$ from the true transmission rate $\beta^*$. (b) Step 2: Epidemiological relevance of observed network with missing data. Each box summarizes log Bayes factor of observed network with missing data compared to null hypothesis (viz a prior of networks with 10% to 100% permuted edges). (c) Evidence for the true synthetic network over datasets with missing data. Log Bayes factor of more than 2.44 (dashed line) is considered to be a strong support in favor of the observed contact network. The middle black line in each box plot is the median, the boxed area extends from the 25th to 75th quartile, and whiskers extended from the hinge to the largest/smallest value no further than 1.5 times the inter-quartile range.

Additionally, removing cases (i.e., infected status from nodes) from the infection time-series results in lower quality infection data compared to data where all infection events are documented.

## Comparison with previous approaches

Next, we compared INoDS with two previous approaches that have been used to establish an association between infection spread and contact network in a host population—the $k$-test and network position test. The $k$-test procedure involves estimating the mean infected degree (i.e., number of direct infected contacts) of each infected individual in the network, called the $k$-statistic. The $p$-value in the $k$-test is calculated by comparing the observed $k$-statistic to a distribution of null $k$-statistics which is generated by randomizing the node-labels of infection cases in the network [31]. Network position test compares the degree of infected individuals to that of uninfected individuals [24, 25, 27]. The observed network is considered to be epidemiologically relevant when the difference in average degree between infected and uninfected individuals exceeds the degree difference in an ensemble of randomized networks at 5% significance level. Both these previous approaches only provide evidence for static networks by comparison with a null expectation. We therefore performed comparisons with step 2 of INoDS where epidemiological relevance of a network hypothesis is evaluated. To do so, we performed simulations of infection with $\beta$ ranging from 0.01 to 0.1, corresponding to disease prevalence of 8.1% to 99.8%, respectively.

We found that INoDS accurately established epidemiological relevance across a wide range of $\beta$ when no network or disease data was missing. At $\beta = 0.01$, however, the power of the model is lower compared to the $k$-test (Fig 4). For values of $beta$ beyond 0.01, the performance of INoDS in establishing epidemiological relevance surpasses two previous approaches—the $k$-test procedure and the network position test. The power of the $k$-test in detecting epidemiological relevance of an observed contact network decreases with an increasing amount of missing data and transmission rate of pathogen. Of the three approaches, the network position test has the lowest power in detecting epidemiological relevance.

## Applications to empirical data-sets

We next demonstrate the application of INoDS to perform hypothesis testing on contact networks, identify transmission mechanisms and infer transmission rate using two empirical datasets. The first dataset is derived from the study by Otterstatter & Thomson [36] that examines the spread of an intestinal pathogen (*Crithidia bombi*) within colonies of the social bumble bee, *Bombus impatiens*. The second dataset documents the spread of *Salmonella enterica* within two wild populations of Australian sleepy lizards *Tiliqua rugosa* [37]. We chose these two empirical datasets because they represented two distinct (i) host taxonomic class, (ii) models of disease spread (SI vs SIS), and (iii) disease data collection methodology (infection timing known for bumble bee dataset vs unknown for sleepy lizard dataset, and (iv) network connectivity (fully connected in bumble bees vs sparsely connected in sleepy lizards).

**Determining transmission mechanism and the role of contact intensity: Case study of *Crithidia bombi* in bumble bees.** [36] showed that the transmission of gut protozons, *Crithidia bombi*, in bumble bee colonies is associated with the frequency of contacts with infected nest-mates rather than the duration of contacts. The dynamic contact networks in the experiments were fully connected, i.e., all individuals were connected to each other in the network at all time steps. We extended the previous analysis by answering two specific questions: (1) Does the type of contact (frequency vs duration) matter in transmission?, and (2) Do contact intensity (i.e, the edge weights) between individuals contribute to infection transfer? We performed

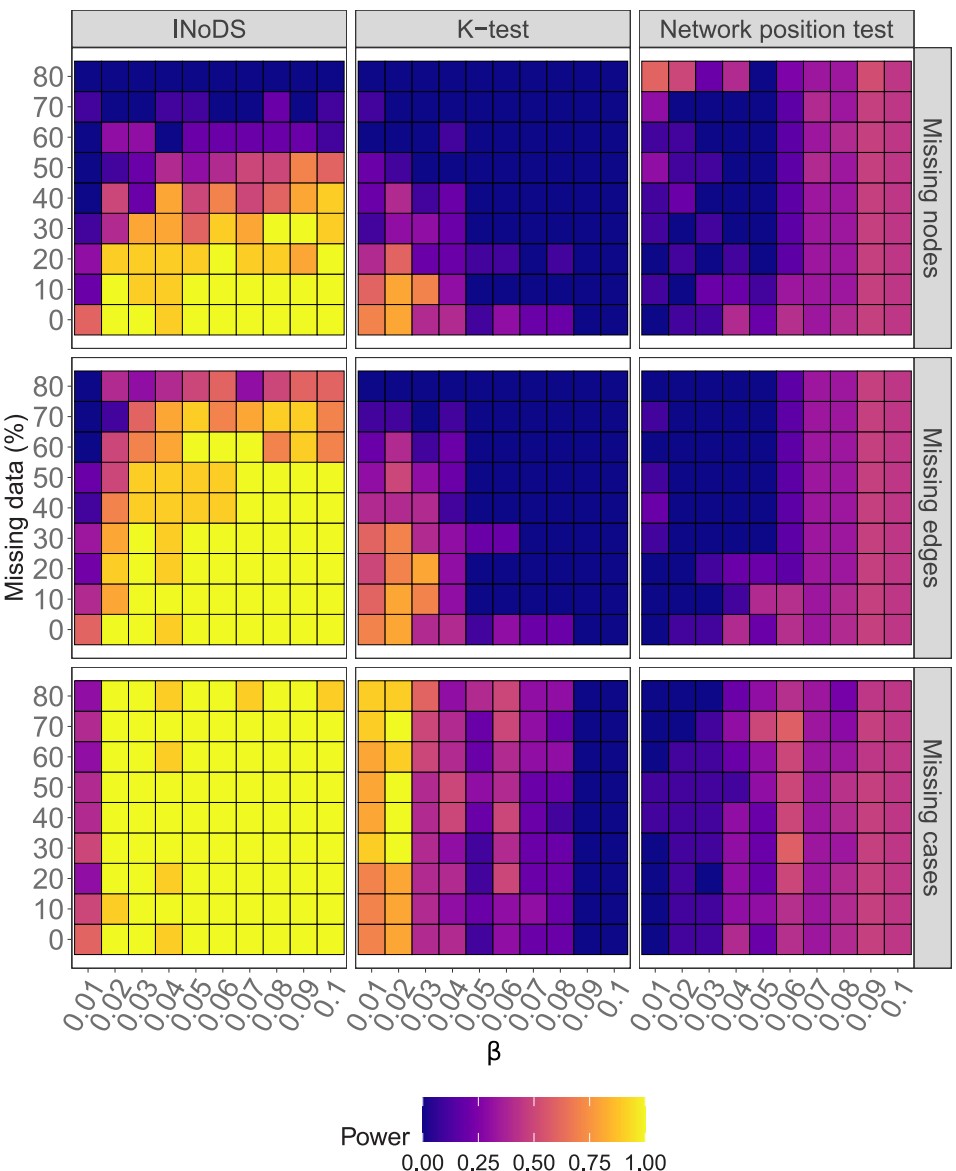

**Fig 4. Comparison of INoDS performance with previous approaches.** Statistical power of INoDS, *k*-test and network position test in establishing epidemiological relevance of the "true" contact network against three common forms of missing data—missing nodes, missing edges and missing infected cases. Statistical power of INoDS, *k*-test and network position test was calculated as the proportion of disease simulations where the observed contact network was detected as epidemiologically relevant (INoDS: $log(B_{10}) > 2.44$; *k*-test and network position test: $p < 0.05$).

analyses on two types of contact network hypotheses—those are described by frequency of contacts and those that are described by duration of contacts—and compared the results with the findings reported in [36].

To answer the two questions, we constructed dynamic contact networks where edges represent close proximity between individuals. Since fully connected networks rarely describe the dynamics of infection spread, we sequentially removed edges with weights less than 5–50% of the highest edge weight to generate contact network hypotheses at different edge weight thresholds. Corresponding to the two types (frequency and duration) of weighted networks,

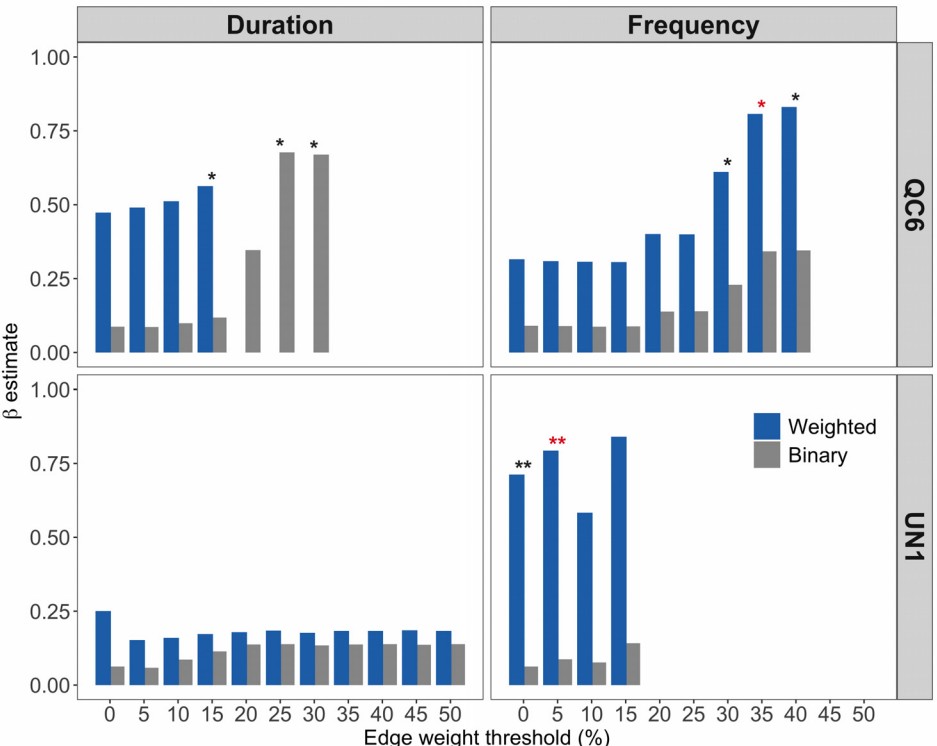

**Fig 5. Identifying the contact network model of *Crithidia* spread in two bumble bee colonies (QC6 and UN1) described in [36].** Edges in the contact network models represent physical interaction between the bees. Since the networks were fully connected, a series of filtered contact networks were constructed by removing weak weighted edges in the network. The x-axis represents the edge weight threshold used to remove weak edges in the network. Two types of edge weights were tested—duration and frequency of contacts. In addition, both types of weighted edges were converted to binary to create binary networks. The results shown are estimated values of the per contact rate transmission rate, $\beta$, for the two colonies. Asterisks above bars indicate that the networks were epidemiologically relevant in explaining the spread of *Crithidia* (single asterisk: $Log(B_{10})$ = 0.5–1, substantial evidence; double asterisks: $Log(B_{10})$ = 1–2, strong evidence). We note that model convergence was not achieved for several network hypotheses and were removed in our final analysis.

unweighted contact networks were also constructed by replacing weighted edges in the thresholded weighted networks with binary edges (i.e., edges with an edge weight of one).

Fig 5 shows the estimates of pathogen transmission rate $\beta$ for the four types of contact network hypotheses at different edge weight thresholds. We found only a few contact network hypotheses were epidemiologically relevant (bars with asterisks). Weighted frequency networks had highest evidence for colonies QC6 and UN1 and at 35% and 5% edge-weight threshold respectively (bars with red asterisks). Our results therefore show that contact frequency, rather than duration, better explained the spread of the *Crithidia bombi* through bumble bee colonies. We also found that contact intensity, quantified through edge weights, is important in explaining pathogen spread.

**Identifying transmission mechanisms with imperfect disease data: Case study of *Salmonella enterica* Australian sleepy lizards.** Spatial proximity is known to be an important factor in the transmission of *Salmonella enterica* within Australian sleepy lizard populations [37]. However, it is not known whether the transmission risk increases with frequency of proximate encounters between infectious and susceptible lizards. We therefore tested two contact network hypotheses to explain the spread of salmonella at two sites of wild sleepy lizards populations. The first contact network hypothesis placed binary edges between lizards if they were

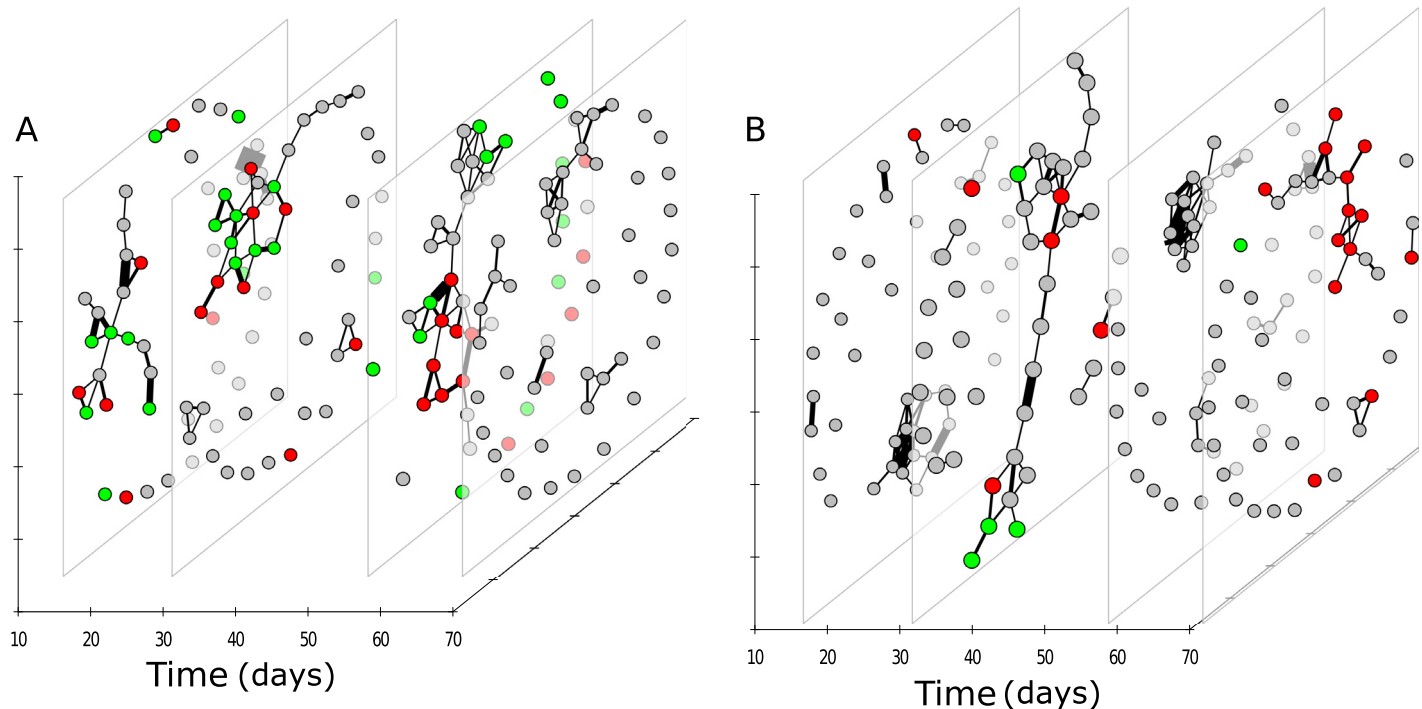

| Site | Transmission mechanism | $\beta$ | $\epsilon$ | Epidemiological relevance | Evidence |
|---|---|---|---|---|---|
| Site 1 | Binary | 0.017 [0.017,0.020] | 0.046 [0.046,0.058] | -0.21 | -169.09 |
| Site 1 | Frequency weighted * | 0.313 [0.020,0.683] | 0.042 [0.031,0.059] | 0.03 | -164.43 |
| Site 2 | Binary | 0.062 [0.034,0.063] | 0.051 [0.051,0.059 | -0.15 | -234.63 |
| Site 2 | Frequency weighted* | 1.761 [0.869,3.684] | 0.049 [0.033,0.068] | **3.34** | -229.57 |

**Fig 6. Identifying transmission mechanisms of Salmonella spread in Australian sleepy lizards.** Dynamic network of proximity interactions for a total duration of 70 days between (A) 43 lizards at site 1, and (B) 44 lizards at site 2. Each temporal slice summarizes interactions within a day (24 hours). Edges indicate that the pair of individuals were within 14m distance of each other, and the edge weights are proportional to the frequency of physical interactions between the node pair. For ease in visualization, four networks summarizing interactions at day 15, 30, 57 and 70 are shown out of a total of 70 static network snapshots. Green nodes are the animals that were diagnosed to be uninfected at that time-point, red are the animals that were diagnosis to be infected and grey nodes are the individuals with unknown infection status at the time-point. We hypothesized that the spatial proximity networks could explain the observed spread of *Salmonella* in the population. The results are summarized as a table. Bold numbers indicate that the network hypothesis was found to be epidemiologically relevant compared to an ensemble of randomized networks. The network hypothesis with the highest log Bayesian (marginal) evidence at each site is marked with an asterisk (*).

ever within 14m distance from each other during a day (24 hours). We constructed the second contact network by assigning edge weights proportional to the number of times two lizards were recorded within 14m distance of each other during a day.

Because disease sampling was performed at regular fortnightly intervals, the true infection time (day) of individuals at both study sites was unknown. We therefore used a data augmentation method in INoDS (see Materials & methods) to sample unobserved infection timings along with the per contact transmission rate, $\beta$, and error, $\epsilon$. We found that weighted network was epidemiologically relevant at site 2 but not at site 1 (Fig 6). Proximity networks with weighted edges had higher marginal (Bayesian) evidence compared with binary networks at both sites. This suggests that the occurrence of repeated contacts between two spatially

proximate individuals, rather than just the presence of contact between individuals is more explanatory of Salmonella transmission in sleepy lizards.

## Discussion

In this study we present INoDS as a tool that performs network model selection and establishes the statistical significance of a contact network model to describe the spread of infectious diseases. Our method also provides epidemiological insights about the host-pathogen system by enabling hypothesis testing on different transmission mechanisms, and estimating pathogen transmission rates. Unlike previous approaches that rely on social network position [24–27], proxy behaviors [28–30] or connectivity [31], we show that our method is robust to missing network data, imperfect disease surveillance, can provide inference for dynamic networks and a range of disease progression models. Additionally, our tool overcomes a common challenge of imperfect knowledge of infection acquisition by assuming infection times to be unobserved and using data on infection *diagnosis* instead to provide inference on contact networks.

In principle, the background transmission rate parameter, $\epsilon$, in INoDS is similar to the asocial learning rate used in the network based diffusion analysis approach in the behavior learning literature [38, 39]. The background transmission parameter in our model serves to approximately assess transmission which is unexplained by the edge connections of the network hypothesis. Under the scenario of a network hypothesis with no edges, the best fit $\epsilon$ would indicate that all disease data is generated by unobserved transmission. Any transmission events that can be better explained by a network edge reduce the model's expectation from this maximum $\epsilon$ estimate. Relative deviation of $\epsilon$ from the maximum possible value is therefore highest for a network hypothesis with no missing data. The relative deviation of $\epsilon$ declines with increasing missing data, although it is more sensitive to missing network information compared with missing information in infection time-series data (S4 Fig).

Our work thus addresses a growing subfield in network epidemiology that leverages statistical tools to infer contact networks using all available host and disease data [9, 31, 40–42]. Our approach can be used to tackle several fundamental challenges in the field of infectious disease modeling [21, 22]. First, INoDS can be used to perform model selection on contact network models that quantify different transmission modes; this approach facilitates the identification of infection-transmitting contacts and does not rely on laboratory experimentation (or subjective expert knowledge). Second, INoDS can be used to establish the statistical significance of proxy measures of contact (such as spatial proximity, home-range overlap or asynchronous refuge use) in cases where data on direct interactions between hosts are limited. Third, INoDS can establish the epidemiological role of edge weights in a contact network by performing model selection of contact networks with similar edge connections but different edge weighting criteria.

In the first empirical example involving the spread of the *Crithidia* gut protozoan in bumble bee colonies, we demonstrate that contact networks weighted with respect to frequency, rather than duration, explain the observed patterns of transmission. Our results therefore support the original finding of the study [36], where individual risk of infection was found to be correlated with contact rate with infected nest-mates. We further found that weak ties below a certain edge weight threshold do not play an important role in infection transfer for this empirical system. In the second empirical dataset, we found that frequency of contacts between closely located lizards allows better, i.e. more consistent, predictions on Salmonella transmission. Our results supports results from a previous study which suggests that the bacterial transmission in Australian sleepy lizards occur between closely located animals [37].

As with all models, the results from INoDS should be considered within the context of model assumptions. First, we assumed that the infection process has no latent period. In the future, disease latency can be incorporated into the model by using a data augmentation technique, similar to what we use for inferring infection times. Second, we assumed that the infectiousness of infected hosts and susceptibility of naive hosts is equal for all individuals in the population. Heterogeneity in infectiousness of infected hosts and the susceptibility of naive hosts can be incorporated as random effects in the model. Third, in the current version of our model, we do not consider re-infection of hosts. Re-infectivity of hosts can be easily incorporated by allowing multiple time-points of infection acquisition in Eq (2).

Our results show that the data-collection efforts should aim to sample as many individuals in the population as possible, since missing nodes have the greatest impact (rather than missing edges) on the statistical significance of network models. Since data-collection for network analysis can be labor-intensive and time-consuming, our approach can be used to make essential decisions on how limited data collection resources should be deployed. For example, under a limited capability of recording real-time interactions between hosts, INoDS can identify the minimum time-resolution required during data collection for a network model with sufficient statistical ability to establish epidemiological relevance. Our approach can also be used to improve targeted disease management and control by identifying high-risk behaviors and super-spreaders of a novel pathogen without relying on intensive transmission experiments that take years to resolve.

## Materials & methods

Here we describe INoDs (Identifying contact Networks of infectious Disease Spread), a computational tool that (*i*) estimates per contact transmission rate ($\beta$) of infectious disease for empirical contact networks, (*ii*) establishes the epidemiological relevance of a contact network by performing Bayesian hypothesis testing, and (*iii*) enables discrimination of competing contact network hypotheses, including those based on pathogen transmission mode, edge weight criteria and data collection techniques. Two types of data are required as input for INoDS— infection time-series data, which include infection diagnoses (coded as 0 = not infected and 1 = infected), and time-step of diagnosis for all available individuals in the population; and an edge-list of a dynamic (or static) contact network. An edge-list format is a list of node pairs (each node pair represents an edge of the network), along with the weight assigned to the interaction, and time-step of interaction, with one node pair per line. The tool can be used for unweighted contact networks—an edge weight of one is assigned to all edges in this case. Time-steps of interactions are not required when analysis is performed on static contact networks. The software, implemented in Python, is platform independent, and is freely available at https://github.com/bansallab/INoDS-model. Empirical datasets used in this study are available at https://doi.org/10.7910/DVN/YAHRDJ.

### INoDS formulation

For a susceptible individual, the potential of acquiring infection at each time-step depends on the per contact transmission rate, the total strength of interactions with its infected neighbors at the previous time-step, and a parameter $\epsilon$ that captures the force of infection that is not explained by the individual's social connections. The probability of receiving infection, $\lambda_i(t)$, of a susceptible individual *i* at time *t* is thus calculated as:

$$\lambda_i(t) = 1 - \exp\{-\beta w_i(t-1) - \epsilon\}, \tag{1}$$

where both $\beta$ and $\epsilon$ parameters are $> 0$; $w_i(t-1)$ denotes the total strength of association

between the focal individual $i$ and its infected associates at the previous time-step $(t - 1)$. For binary (unweighted) contact network models $w_i = k_i$, where $k_i$ is the total infected connections of the focal individual.

The log-likelihood of the observed infection time-series data given the contact network hypothesis $(H_A)$ can therefore be estimated as:

$$\mathcal{L}(D|H, \Theta) = \log(D|H_A, \beta, \epsilon) = \sum^n \log[\lambda_n(t_n)] + \sum^t \left( \sum^m \log[1 - \lambda_m(t)] \right), \qquad (2)$$

where $D$ is the infection time-series data, the set of unknown parameters is $\Theta = \{\Theta_1, \Theta_2. ... \Theta_N\}$, $t_n$ is the time of infection of individual $n$, and t are all time-steps at which individual $n$ is naive and has the ability to contract infection. The first part of Eq 2 estimates the log likelihood of all observed infection acquisition events. The second part of the equation represents the log-likelihood of susceptible individuals $m$ remaining uninfected at time $t$.

Following Bayes' theorem, the posterior distribution of the set of parameters is given as:

$$P(\Theta|D, H) = \frac{\mathcal{L}(D|H, \Theta)\mathcal{P}(\Theta|H)}{\mathcal{M}(D|H)} \propto \mathcal{L}(D|H, \Theta)\mathcal{P}(\Theta|H) \qquad (3)$$

where $D$ is the infection time-series data, $H$ is the contact network hypothesis, and $P, \mathcal{L}, \mathcal{P}, \mathcal{M}$ are the shorthands for the posterior, the likelihood, the prior and the marginal likelihood, respectively.

**Parameter estimation and data augmentation of infection timings.** We used `DynamicNestedSampler` of *dynesty* package implemented in Python to estimate Bayesian posteriors and evidences [43]. Nested sampling is a numerical method of simultaneously estimating both the posterior and evidence by maintaining a set of samples from the prior, and iteratively updating them subject to the constraint that new samples have higher likelihoods. Dynamic nested sampling allows the samples to be allocated adaptively, maximizing both accuracy and efficiency [43]. We assumed a uniform prior distribution for $\beta$ and $\epsilon$ parameters with range [0, 10].

Calculation of the likelihood in Eq 2 requires knowledge of exact timing of infection, $t_1, . . .t_n$, for $n$ infected individuals in the population. However in many cases, the only data available are the timings of when individuals in a populations were *diagnosed* to be infected, $d_1, . . .d_n$. We therefore employ a Bayesian data augmentation approach to estimate the actual infection timings in the disease dataset [44]. Since in this case the infection time $t_i$ for an individual $i$ is unobserved, we only know that $t_i$ lies between the interval $(L_i, d_i]$, where $L_i$ is the last negative diagnosis of individual $i$ before infection acquisition. Within this interval, the individual could have potentially acquired infection at any time-step where it was in contact with other individuals in the network. Assuming incubation period to be one time-step, the potential set of infection timing can be represented as $t_i \in \{g_i(t_i - 1) > 0, L_i < t_i \leq d_i\}$, where $g_i(t_i - 1)$ is the degree (number of contacts) of individual $i$ at time $t_i - 1$. For infections that follow a SIS or SIR disease model, it is also essential to impute the recovery time of infected individuals for accurate estimation of infected degree. To do so, we adopt a similar data augmentation approach as described to sample from the set of possible recovery time-points.

The data augmentation proceeds in two steps. In the first step, the missing infection times are imputed conditional on the possible set of infection times. In the next step the posterior distributions of the unknown parameters are sampled based on the imputed data. We performed data imputation using inverse transform sampling method, which is a technique of drawing random samples from any probability distribution given its cumulative distribution function [45].

**Interpretation of the ε parameter.**   In principle, inclusion of the $\epsilon$ parameter in Eq 1 is similar to the asocial learning rate used in the network based diffusion analysis approach in the behavior learning literature [38, 39]. The background transmission parameter $\epsilon$ in INoDS formulation serves to approximately assess transmission which is unexplained by the edge connections of the network hypothesis. Estimate value of $\epsilon$ parameter increases with missing data, although it is more sensitive to missing network information compared with missing information in infection time-series data (S3 Fig).

**Epidemiological relevance of a contact network.**   We performed Bayesian hypothesis testing to establish the epidemiological relevance of a contact network in explaining the observed pattern of infectious disease spread. Null hypothesis is expressed as a uniform prior distribution over networks with permuted edge connections of varying degree (10% to 100%). Prior distribution of alternative hypothesis puts all credibility in an infinitely dense spike at 0% permutation level (viz, the observed contact network). Bayes factor of alternate hypothesis vs null was defined as:

$$BF_{alt} = \frac{p(M = alt|D)}{p(M = null|D)} \frac{p(M = null)}{p(M = alt)} \tag{4}$$

Log Bayes factor was calculated as the difference between log marginal likelihoods of the alternate and null hypothesis. A log Bayes factor of 0.5 –1, 1–2 and >2.44 is considered as substantial, strong and decisive evidence, respectively, towards the epidemiological relevance of the observed contact network [46, 47].

**Model selection of competing network hypotheses.**   To facilitate model selection in cases where there are more than one network hypothesis, we compute marginal likelihood of the infection data given each contact network model. The marginal likelihood, also called the Bayesian evidence, measures the model fit, i.e,. to what extent the infection time-series data can be simulated by a network hypothesis ($H_1$). Bayesian evidence is based on the average model fit, and calculated by integrating the model fit over the entire parameter space.

$$P(D|H, \Theta) = \int \mathcal{P}(\Theta|H)\mathcal{L}(D|H, \Theta)d\Theta \tag{5}$$

Dynamic nested sampling calculates the evidence by integrating the prior within nested contours of constant likelihood. Model selection can be then performed by computing pair-wise Bayes factor, i.e. the ratio of the marginal likelihoods of two network hypotheses. The log Bayes factor to assess the performance of network hypothesis $H_A$ over network hypothesis $H_B$, is expressed as:

$$log(BF_{BA}) = log(P(D|H_B)) - log(P(D|H_A)) \tag{6}$$

The contact network with a higher marginal likelihood is considered to be more plausible, and a log Bayes' factor of more than 2.44 is considered to be a decisive support in favor of the alternative network model ($H_B$) [46, 47].

## Validating INoDS performance

We validated the performance of INoDS by evaluating its accuracy in estimating the unknown transmission parameter $\beta$, robustness to missing data, and comparing the performance of tool with previous approaches. To do so we first constructed a dynamic synthetic network using the following procedure. At time-step $t = 0$, a static network of 100 nodes, mean degree 4, and Poisson degree distribution was generated using the configuration model [35]. At each subsequent time-step, 10% of edge-connections present in the previous time-step were permuted,

for a total of 100 time-steps. That is, the dynamic synthetic network (called true synthetic network in the results) consists of contacts $(i, j, t)$ between node $i$ and $j$ at time $t$, lasting from a duration of 1 to 100 time steps; each contact is capable of disease transmission [48]. Next, through the synthetic dynamic network, we performed 10 independent SI disease simulations with per contact rate of infection transmission ($\beta^*$) 0.01 to 0.1. Model accuracy was evaluating by comparing INoDS estimation of the transmission parameter, $\beta$, with the true transmission rate $\beta^*$ that was used to perform disease simulations. Since the synthetic network completely described the disease simulations, model accuracy was also tested by evaluating the deviation of the estimated error parameter $\epsilon$, from the expected value of zero.

We next tested robustness of the tool against two types of missing network data: (a) incomplete sampling of individuals in a population (missing nodes), and (b) incomplete sampling of interactions between individuals (missing edges). To do so, we simulated two independent SI disease simulations each for ten $\beta^*$ values ranging from 0.01 to 0.1 with increment of 0.01 through the synthetic dynamic network. The two scenarios of missing data were created by randomly removing 25–75% of nodes and edges from the true synthetic network that were not a part of the path of simulated infection spread. This approach allowed us to investigate INoDS performance for various levels of incompletely sampled networks with the infection path preserved. If INoDS is robust to missing network data, we expect to recover similar parameter estimate, epidemiological relevance and model evidence for all networks.

## Comparisons with previous approaches

We compared INoDS performance with two previous approaches that are popularly used to establish epidemiological significance of contact networks—$k$-test [31] and network position test. $k − test$ estimates the average number of infected connections for each infected individual in the network. To establish the epidemiological significance of a contact network this metric, called the $k$-statistic, is compared to a distribution of null $k$-statistics obtained by randomly swapping the node labels. A p-value is calculated as the number of permutations that produce $k$-statistics more extreme than the observed $k$-statistic. Network position test compares the average degree (i.e., the average number of connections) of infected cases with the degree of uninfected individuals. The difference in average degree in the observed network is compared to the degree difference in networks where random edge connections. A p-value is calculated as the number of random networks where the degree between infected and uninfected individuals is higher than the observed network.

Both of these previous approaches only provide epidemiological evidence of the observed contact network and do not estimate transmission parameters or enable model comparisons. We therefore performed comparisons with step 2 of INoDS where epidemiological relevance of the observed network is evaluated. We also note that the previous approaches are limited to static and unweighted networks, we therefore performed model comparisons on a "true" synthetic static network with 100 nodes, Poisson degree distribution and a mean network degree of 3. Simulations of disease spread were performed with a broad range of per contact transmission rate ($\beta$). Null expectation in INoDS and network position test was generated by permuting the edge connections of the observed networks, creating an ensemble of null networks. In $k$-test, the location of infection cases within the observed network are permuted, creating a permuted distribution of $k$-statistic [31].

## Applications to empirical datasets

We demonstrate the applications of our approach using two datasets from the empirical literature. The first dataset comprises of dynamic networks of bee colonies (N = 5–7 individuals),

where edges represent direct physical contacts that were recorded using a color-based video tracking software. A bumble bee colony consists of a single queen bee and infertile workers. Here, we focus on the infection experiments in two colonies (colony QC6 and UN1). Infection progression through the colonies was tracked by daily screening of individual feces, and the infection timing was determined using the knowledge of the rate of replication of *C. bombi* within its host intestine.

The second dataset monitors the spread of the commensal bacterium *Salmonella enterica* in two separate wild populations of the Australian sleepy lizard *Tiliqua rugosa*. The two sites consisted of 43 and 44 individuals respectively, and these represented the vast majority of all resident individuals at the two sites (i.e., no other individuals were encountered during the study period). Individuals were fitted with GPS loggers and their locations were recorded every 10 minutes for 70 days. Salmonella infections were monitored using cloacal swabs on each animal once every 14 days. Consequently, the disease data in this system do not identify the onset of each individual's infection. We used a SIS (susceptible-infected-susceptible) disease model to reflect the fact that sleepy lizards can be reinfected with salmonella infections. Proximity networks were constructed by assuming a contact between individuals whenever the location of two lizards was recorded to be within 14m distance of each other [26]. The dynamic networks at both sites consisted of 70 static snapshots, with each snapshot summarizing a day of interactions between the lizards. We constructed two contact network hypotheses to explain the spread of salmonella. The first contact network hypothesis placed binary edges between lizards if they were ever within 14m distance from each other during a day. The second contact network assigned edge weights proportional to the number of times two lizards were recorded within 14m distance of each other during a day. Specifically, edge weights between two lizards were equal to their frequency of contacts during a day normalized by the maximum edge weight observed in the dynamic network.

## Supporting information

**S1 Fig. Corner plot of the posterior sample showing relationship between estimates of $\beta$ and $\epsilon$.** True value of the parameters are indicated with red lines. Contours contain 25%, 50% and 75% of the sample points respectively.
(EPS)

**S2 Fig. Estimate of $\epsilon$ with increasing percentages of nodes/edges removal that were not involved in infection spread path.**
(EPS)

**S3 Fig. Robustness of INoDS to random missing data. Networks with missing data were created by randomly removing 10–60% of nodes, edges and cases.** (a) Step 1: $\Delta\beta$ is the deviation of estimated transmission parameter $\beta$ from the true transmission rate $\beta^*$. (b) Step 2: Epidemiological relevance of observed network with missing data. Each box summarizes log Bayes factor of observed network with missing data compared to null hypothesis (viz a prior of networks with 10% to 100% permuted edges). (c) Evidence for the true synthetic network over datasets with missing data. Log Bayes factor of more than 2.44 (dashed line) is considered to be a strong support in favor of the observed contact network. The middle black line in each box plot is the median, the boxed area extends from the 25th to 75th quartile, and whiskers extended from the hinge to the largest/smallest value no further than 1.5 times the inter-quartile range.
(EPS)

**S4 Fig. Relative deviation of estimated $\epsilon$ from the maximum possible value with increasing percentage of random missing data.** Maximum $\epsilon$ was determined by maximizing the likelihood function in Eq (2) assuming network with no edges. Each boxplot summarizes the results of 20 independent SI disease simulations through a synthetic dynamic network—two each for ten $\beta^*$ values ranging from 0.01 to 0.1 with increment of 0.01. Missing data were created by randomly removing 10–60% of nodes, edges or cases from the dataset. The middle black line in each box plot is the median, the boxed area extends from the 25th to 75th quartile, and whiskers extended from the hinge to the largest/smallest value no further than 1.5 times the interquartile range.
(EPS)

**S5 Fig. Disease prevalence corresponding to different values of $\beta$ used in our simulations for Fig 4.**
(EPS)

## Acknowledgments

We thank Peter Majoros, Tom Haley, Alienor Quiblier and Ben Westwood for their assistance with lizard fieldwork, and David Gordon for his lab work.

## Author Contributions

**Conceptualization:** Pratha Sah, Shweta Bansal.

**Data curation:** Pratha Sah, Michael Otterstatter, Stephan T. Leu.

**Formal analysis:** Pratha Sah.

**Funding acquisition:** Shweta Bansal.

**Investigation:** Pratha Sah, Shweta Bansal.

**Methodology:** Pratha Sah, Sivan Leviyang, Shweta Bansal.

**Project administration:** Shweta Bansal.

**Resources:** Michael Otterstatter, Stephan T. Leu, Shweta Bansal.

**Software:** Pratha Sah, Shweta Bansal.

**Supervision:** Shweta Bansal.

**Validation:** Pratha Sah, Stephan T. Leu, Sivan Leviyang, Shweta Bansal.

**Visualization:** Pratha Sah.

**Writing – original draft:** Pratha Sah.

**Writing – review & editing:** Pratha Sah, Michael Otterstatter, Stephan T. Leu, Sivan Leviyang, Shweta Bansal.

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
