## [Decision Letter · Decision Letter 0]

18 Jan 2021

Dear Dr. Sah,

Thank you very much for submitting your manuscript "Revealing mechanisms of infectious disease outbreak through empirical contact networks" for consideration at PLOS Computational Biology.

As with all papers reviewed by the journal, your manuscript was reviewed by members of the editorial board and by several independent reviewers. In light of the reviews (below this email), we would like to invite the resubmission of a significantly-revised version that takes into account the reviewers' comments.

We cannot make any decision about publication until we have seen the revised manuscript and your response to the reviewers' comments. Your revised manuscript is also likely to be sent to reviewers for further evaluation.

Sincerely,

Benjamin Muir Althouse

Associate Editor

PLOS Computational Biology

Rob De Boer

Deputy Editor

PLOS Computational Biology

Reviewer's Responses to Questions

**Comments to the Authors:**

Reviewer #1: This manuscript presents a method of inferring model parameters for disease spreading on empirical contact networks. It is also capable of testing the robustness of the network representations given data about who is infected at what time. I think it looks like a contribution of practical importance, even though getting the data needed for input is a difficult task. Below are some specific concerns:

[Line 9–12] "Constructing a complete contact network model requires (i) knowledge about the transmission route(s) of a pathogen, (ii) a sampling of all individuals in a population, and (iii) a sampling of all interactions among the sampled individuals that may lead to infection transfer." First, if you have done (iii), haven't you also done (ii)? Second, why do you need to know (i)? Yes, you would need to know it to infer the parameters of the compartmental model, but not to construct a contact network?

[Line 61–65] You say you can use a "dynamic contact network." Above, you defined a contact network as "individuals are represented as nodes, and an edge between two nodes represents an interaction that has the potential to transmit infection". Exactly how does dynamism enter this definition?

[Line 62–63] If you have a temporal network version about the SIS or SIR model, it is a two-parameter model. You cannot reduce parameters and omit the recovery rate as in static network epidemiology. Thus, your tool also needs to estimate the recovery rate (or disease duration, depending on your model of infection duration).

[Line 63] In the light of the previous question, you can omit the SI model since it is contained in the SIS and SIR models.

[Fig. 1] The smallest text is not readable.

[Around line 87] How do you randomize the network? If you randomize a dynamical network, there are many options that all destroy different structures of the original data. See Gauvin et al., Randomized reference models for temporal networks

https://arxiv.org/abs/1806.04032

[Around line 87] What is the logic of randomizing different percentages of the links? When you randomize the network, you destroy the particular structure that affects the disease spreading. It is not telling you whether your network is robust to errors because random changes make the network more random, i.e., less like the structured, real networks. If you consider networks obtained by random changes as a model, this is systematically biased toward randomness and should certainly be a bad model.

[Line 104] " 0.039 ( 0.003 SD)" What is this notation? Do you mean 0.003 times the standard deviation? or that 0.003 is the standard deviation?

[Line 118] What is an "edge" in a dynamic network? Please conform to the temporal-network literarure. Masuda & Holme, Predicting and controlling infectious disease epidemics using temporal networks, F1000Prime Rep. 5, 6 (2013).

Continuing the previous point. There are papers in the temporal-network epidemiology literature that relate to this work. One such is: Holme & Rocha, Impact of misinformation in temporal network epidemiology, Network Science 7, 52-69 (2019).

[Line 122] How do you define "true synthetic network"?

[Fig. 3] What is the point of plotting panel A? You could just state what we see in a sentence.

[Fig. 3 caption] If you remove edges not involved in a spreading path, aren't you introducing a bias? Those edges should, by definition, be less important for the epidemics.

Reviewer #2: Sah, et al. present an exciting technique that is leaps and bounds beyond of the "comparable" approaches they test it against in the study of contact network relevance to disease spread. This work has great potential to address some of the most pressing concerns in disease ecology, namely a rigorous way to assess and compare contact networks as explanatory tools in the spread of infectious disease. Despite my excitement and generally positive impression, I have one main concern and two secondary ones that I outline below.

# Validation with respect to missing data:

The authors perform three main tests to validate their approach in the face of missing data: they remove nodes, edges, or cases and then run the model to see if it still performs well in each of its three tasks: estimating parameters, determining that the network is relevant to disease spread, and performing model selection between the "true network" and the pruned ones.

My main issue is that under the current methods, the authors only consider nodes and edges that are not involved in the simulated disease transmission for removal. Besides being unrealistic from a data-scarcity perspective (we rarely know whether the individuals we have missed were involved in the transmission chain or not), this additionally (I would argue undesirably) alters the model-perceived disease prevalence.

Because (by definition) none of the removed edges were involved in the transmission process, many (most?) of them were connecting two susceptible nodes and thus wouldn't have come into the likelihood calculation anyway. The removal of edges should functionally increase the perception of beta, since edges that remain are more likely to result in further infections, but this is not what we see in Figure 3 -- I wonder if the authors have an intuition for why their model continues to perform so well

The removal of nodes is not really a distinct treatment from the removal of edges, as when nodes are removed, the edges connected to them are removed as well. We would thus expect nodes to have (at least) as big of an effect as just removing edges. Importantly, this removal also increases the prevalence of disease within the network (because we are only removing healthy nodes). This combined with the above is likely why we see an increase in epsilon, though the lack of a response in beta is surprising, as before.

Finally, removing cases is the only change that alters the actual infection data, but in this case, nodes are not removed but rather their infectious status is never recognized (assuming they follow the procedure from VanderWaal, et al. (2016)). This seems like it would have the biggest effect on beta, but those results are not presented in this work. It does yield the smallest absolute epsilon, but might have something to do with effectively reducing prevalence in the population. Thus, less extrinsic forcing is needed to explain the data, even in the absence of beta (more on this below).

I would like to see two additional pruning procedures: removing nodes truly at random (infected or not), and the same for edges. This would keep the prevalence approximately equal and would be a more realistic scenario. I expect these will have a larger effect on beta, since we are losing information about the actual spread of disease.

It might be interesting to connect the neighbors of removed nodes via the addition of edges, since there is still a path for infection even if we don't observe it, but this might be beyond the scope of this work.

In Figure S2, the authors consider the absolute deviation of epsilon from 0, but I am not convinced this is the most informative way to present the model's reliance on unobserved transmission. Given a timeseries and a network without edges, there is some best-fit (i.e. maximizing the likelihood function in equation 2) epsilon for regenerating the data. Any transmission events that can be better explained by a network edge (and therefore beta parameter) reduce the model's expectation for epsilon. I am curious how a figure like S2 would look if the y-axis were the relative error (to this maximum) rather than absolute error (from 0).

Finally, my understanding is that the time of infection is assigned (uniformly) randomly between the two timepoints of an individual being susceptible and infectious. It seems a reasonable assumption that this distribution is non-uniform insofar as the likelihood of infection will depend on the node's degree as it changes throughout this window.

# Comparison to previous methods:

One of the variables noted to be important to success of the approach of VanderWaal, et al. (2016) is pathogen prevalence. Did the authors detect a similar dependence in their method?

Along these lines, I am having trouble reconciling Figure 4 with Figure 3 from VanderWaal, et al. (2016) -- how do the authors find nearly 0 power under (presumably) similar parameterizations that VanderWaal, et al. find nearly 1 (i.e. beta = 0.04 and 0.133)?

l 378 -- the authors use the Kass & Rafferty (1995) interpretations of BF (i.e. > 0.5 is "signficant"), but this seemed to be the most generous of interpretations that I've encountered. In most cases, this doesn't matter, as the authors report the BF directly, however in Figure 3, I wonder how much the left column would differ if 2, 3, or 10 (sensu Lee and Wagenmakers (2014)) were used instead.

# Code/reproducibility:

The link provided in the text (https://bansallab.github.io/INoDS-model/) leads to a page that does not contain a link to download the software, requiring the reader to have sufficient knowledge of github to navigate to https://github.com/bansallab/INoDS-model/ before gaining access.

It would additionally greatly improve accessibility if some sample data were provided in the repository such that a user can get it running right out-of-the-box, so to speak. As it is, even the referenced datasets are not readily available. I was unable to find either dataset within the Harvard Dataverse (as referenced in the text), and could find no record of the bee dataset elsewhere either. I did find the lizard dataset in DataDryad (https://datadryad.org/stash/dataset/doi:10.5061/dryad.jk87h), but the data format in this repository is not easily converted into one acceptable for INoDS.

I don't think the randomization code used in this work was provided -- does your particular configuration model algorithm allow non-simple graphs? If so, I would recommend re-running with one that does not in order to make the random ensemble representative of empirically obtainable data.

# Other Minor Points

My read of Eq. 2 is that it this formulation is particular to a disease model without re-infection -- if so, this should be stated explicitly

l 136-138 -- I'm not sure I understand why this was "expected"

l 157 -- this reference should be to figure 4, I believe

l 362 -- should this section heading have an "epsilon" in it?

l 411 -- do I understand these methods correctly that there was only one simulation per missing data scenario-beta combination? If so, I assume the paired outliers in Figure 3c are thus corresponding to particular beta values. I highly recommend performing greater replication in this analysis.

## Figure 2

Contrary to the caption, the boxplot whiskers do not encompass the range (as evidenced by the presence of points outside of these ranges). More likely:

"The upper whisker extends from the hinge to the largest value no further than 1.5 * IQR from the hinge (where IQR is the inter-quartile range, or distance between the first and third quartiles). The lower whisker extends from the hinge to the smallest value at most 1.5 * IQR of the hinge. Data beyond the end of the whiskers are called "outlying" points and are plotted individually." -ggplot2 geom_histogram help page

This description is also present in other captions.

a) Plotting the beta/epsilon values directly here makes it somewhat difficult to gauge any deviation of the mean estimate from the ground truth, or any trends in deviation along beta*. Perhaps switch the vertical axis to show the absolute error or add a 1:1 line for reference. I slightly prefer the former approach, since that would allow a better understanding of any change in epsilon with increasing beta as well, which is hard to note in the current figure due to the difference in scales.

b) Do these boxplots aggregate log BF across networks that vary in their degree of permutation (i.e. combining what is subdivided in panel c)?

## Figure 3

I wasn't sure why omitted cases is not presented here, despite being in Figure 4 and in the SI

a) Are the differences from expectation actually 0 or is this a vertical-axis scaling issue? and is this absolute error? if so, I would recommend relative error here, since beta spans an order of magnitude.

## Figure 4

caption -- "Statistical power of INoDS was ," [delete was]

## Figure 6

Is there a missing network in this figure? I expected 5 networks given biweekly sampling for 70 days.

I'm a little confused by the table here -- shouldn't frequency weighted at site 2 have an asterisk after it? and why does it have "frequency" in parentheses? Finally, am I reading this correctly insofar as the lower CI bound for the beta and epsilon of the site 1 binary network are equal to the estimates themselves (0.017 and 0.046, respectively)?

Reviewer #3: This is an interesting and well written paper describing the development of a tool, INoDS (Identifying contact Networks of infectious Disease Spread), that can be used to perform network model selection and establishes the statistical significance of a contact network model.

General comments:

This work is very timely. The introduction, methods, and results are well described and easy to follow. However, there are few key components in the methods and results that need more explanation. Something that the authors need to explain in more detail is why the bumble bee and lizard case studies were chosen and why there was a need to examine both and not just one, for example. Additionally, while the introduction, methods, and results were well developed, the discussion is currently a bit weak, as it essentially re-iterates much of the information given in the abstract, introduction, and results. The authors do not discuss their work in the context of other studies (or do so at a very shallow level), there is no discussion of limitations nor of future directions and next steps. I would suggest cutting much of the information that repeats the introduction (e.g., why this tool is needed) and results, and provide a deeper discussion of how this tool should be used, which systems might need it most, and what the limitations are. Finally, the authors need to check some of their references and figure citations in the main text before publication as there are errors (e.g., some figures are incorrectly referenced in the main text (e.g., figure 3 instead of 4) and figure 6 was not referred to at all in the main text).

Specific comments:

Line 63-64: provide examples in parentheses of the type of complex models

Line 94-95: More detail on why those two empirical datasets were chosen, and why the need to look at both. This can be stated either here or in the methods.

Line 157: I believe the authors are referring to figure 4 here not figure 3.

Line 292-293: In what way? Perhaps give examples.

Line 458-459: This needs to be stated earlier in the manuscript as it helps understand why this dataset was used as an example.

Figures:

Figure 1 – In the first inferential step, need to define k. Since the methods are at the end of the manuscript, it would be good to define it here as well. In the third inferential step, might be good to explain that a Bayes factor above 3 provides support for one hypothesis over the other (for those less familiar with Bayesian statistics).

Figure 3 – panels a-c are not listed in the figure but are when the figure is referenced in the text.

Figure 6 – not referred to in main text

Figure 6 – suggest adding days in parentheses for the time on the x-axis.

**Have all data underlying the figures and results presented in the manuscript been provided?**

Reviewer #1: Yes

Reviewer #2: **No: **See full comments for details

Reviewer #3: None

PLOS authors have the option to publish the peer review history of their article (what does this mean?). If published, this will include your full peer review and any attached files.

Reviewer #1: No

Reviewer #2: **Yes: **Matthew J Michalska-Smith

Reviewer #3: No
---

## [Decision Letter · Decision Letter 1]

30 Jul 2021

Dear Dr Bansal,

Thank you very much for submitting your manuscript "Revealing mechanisms of infectious disease outbreak through empirical contact networks" for consideration at PLOS Computational Biology. As with all papers reviewed by the journal, your manuscript was reviewed by members of the editorial board and by several independent reviewers. The reviewers appreciated the attention to an important topic. Based on the reviews, we are likely to accept this manuscript for publication, providing that you modify the manuscript according to the review recommendations.

Sincerely,

Benjamin Muir Althouse

Associate Editor

PLOS Computational Biology

Rob De Boer

Deputy Editor

PLOS Computational Biology

[LINK]

Reviewer's Responses to Questions

**Comments to the Authors:**

Reviewer #2: I thank the authors for their work. They have comprehensively addressed all of my major concerns.

Two minor lingering comments:

1) The axes in figure 2a could be better fit to the data (right now the data only fill the lower ~1/3 of the plot.

2) As noted in my original review, the replication count for figure 3 feels quite small (only 2 replicates per parameter combination) -- while I would have liked to see more, this is not a sticking point for my endorsement

**Have the authors made all data and (if applicable) computational code underlying the findings in their manuscript fully available?**

Reviewer #2: Yes

PLOS authors have the option to publish the peer review history of their article (what does this mean?). If published, this will include your full peer review and any attached files.

Reviewer #2: No

Figure Files:

Data Requirements:

Reproducibility:

References:

---

## [Editor Report · Decision Letter 2]

31 Oct 2021

Dear Dr Bansal,

We are pleased to inform you that your manuscript 'Revealing mechanisms of infectious disease outbreak through empirical contact networks' has been provisionally accepted for publication in PLOS Computational Biology.

Best regards,

Benjamin Muir Althouse

Associate Editor

PLOS Computational Biology

Rob De Boer

Deputy Editor

PLOS Computational Biology

---

## [Editor Report · Acceptance letter]

8 Dec 2021

PCOMPBIOL-D-20-02017R2 

Revealing mechanisms of infectious disease spread through empirical contact networks

Dear Dr Bansal,

I am pleased to inform you that your manuscript has been formally accepted for publication in PLOS Computational Biology. Your manuscript is now with our production department and you will be notified of the publication date in due course.

With kind regards,

Olena Szabo
